# Clinical and Metabolic Particularities of a Roma Population with Diabetes—Considering Ethnic Disparities in Approaching Healthcare Management

**DOI:** 10.3390/biomedicines12071422

**Published:** 2024-06-26

**Authors:** Andrada Cosoreanu, Emilia Rusu, Florin Rusu, Silviu Stanciu, Ioana Ungureanu, Marius Donici, Alexandra Visinescu, Georgiana Enache, Gabriela Radulian

**Affiliations:** 1Department of Diabetes, Nutrition and Metabolic Diseases, “Carol Davila” University of Medicine and Pharmacy, 050474 Bucharest, Romania; andrada.cosoreanu@drd.umfcd.ro (A.C.); alexandra-mihaela.visinescu@drd.umfcd.ro (A.V.); gabriela.radulian@umfcd.ro (G.R.); 2“Doctor Carol Davila” Central Military University Emergency Hospital, 010825 Bucharest, Romania; florinrusumd@yahoo.com (F.R.); silviu.stanciu@umfcd.ro (S.S.); 3“Nicolae Malaxa” Clinica Hospital, 022441 Bucharest, Romania; ioana.ungureanu97@yahoo.com (I.U.); mdonici@yahoo.com (M.D.); 4“Pompei Samarian” Emergency Hospital, 910071 Calarasi, Romania; georgianamd@yahoo.com

**Keywords:** clinical characteristics, metabolic particularities, anthropometric assessment, Roma population

## Abstract

The Roma population is Europe’s largest ethnic minority, yet data on the prevalence of non-communicable diseases remain scarce in medical literature. This study aimed to compare the clinical and metabolic particularities of a Roma population with diabetes with a group of non-Roma. We conducted an observational, transversal study and evaluated 808 adult patients with diabetes mellitus, from a tertiary diabetes care hospital. The prevalence of metabolic syndrome was high among both groups, 94.3% in the Roma patients and 89.1% in the non-Roma. A slightly higher mean value of the triglyceride–glucose (TyG) index was observed among the Roma group (10.07 ± 0.71 versus 9.71 ± 0.82). Among the non-Roma, variables that were significantly associated with the TyG index were glycated hemoglobin (HbA1c), total cholesterol (TC), high density lipoprotein–cholesterol (HDL-c), and low-density lipoprotein-cholesterol (LDL-c), while among the Roma, HbA1c and HDL-c were correlated with this index. There were no differences concerning myocardial infarction; however, the number of patients with a history of stroke was 2.1 times higher in the Roma group compared to the non-Roma group. The prevalence of cardiovascular risk factors, cardiovascular disease, and microvascular complications among the study’s Roma population are quite significant, underscoring the importance of ethnic disparities in approaching healthcare management strategies.

## 1. Introduction

The Roma population constitutes one of the largest ethnic minorities in Europe, estimated at 10–12 million. In countries with significant Roma populations (Southern, Central, and Eastern Europe), collecting statistical data on health and its determinants based on ethnic status remains challenging [1].

While the connection between extremely adverse socio-economic conditions and the poor health status among Roma is quite clear, the discrepancies noted in comparisons between the Roma population and the general population do not seem to be entirely explained by their inferior socio-economic status. Recent research into the genetic basis of the high risk of various non-communicable diseases among Roma additionally supports the fact that their health status is influenced by a mix of different health-related factors [1].

Studies have highlighted the elevated prevalence of both communicable and non-communicable diseases among the Roma population. Their health and living standards in Central and Eastern Europe are reported to be lower compared to the general population. Studies suggest a higher incidence of diseases like diabetes, coronary heart disease, and obesity within this minority group. Social determinants such as poverty, limited healthcare access, inadequate nutrition, and cultural traditions play significant roles in these disparities. Additionally, due to frequent consanguineous marriages within the Roma community, there is a heightened risk of genetically inherited illnesses emerging among them [1,2]. Therefore, research also indicates a reduced life expectancy among Roma compared to the majority population, their population pyramid typically showing a higher proportion of younger individuals and a lower concentration of elderly individuals, suggesting a predominantly progressive demographic trend [3]. 

Moreover, an important marker in understanding metabolic health disparities is the triglyceride–glucose (TyG) index, which correlates with an increased risk of developing metabolic syndrome, insulin resistance, and cardiovascular events. The TyG index has proven to be an useful marker for early identification of and intervention in metabolic health issues [4].

For many years, the Roma population has been notably overlooked in health policy and research, with only recent developments beginning to address this issue. In response to emerging data concerning the health disparities faced by the Roma, the European Commission and other international entities have advocated the necessity of measures to narrow the health gap between the Roma and the broader population [1].

Therefore, the aim of this study was to compare the clinical and metabolic profiles of Roma and non-Roma patients with diabetes in a tertiary care center in Bucharest, Romania. Specifically, we assessed the prevalence of metabolic syndrome, obesity, hypertension, dyslipidemia, cardiovascular disease, and health-related behaviors (smoking and alcohol consumption), along with anthropometric and paraclinical measurements in both groups.

## 2. Materials and Methods

### 2.1. Study Design and Setting

We conducted an observational, transversal study in October 2022 and March 2024. We evaluated 808 adult patients with diabetes mellitus, aged between 18 and 89 years old who were attending a tertiary hospital providing diabetes care, “Nicolae Malaxa” Clinical Hospital in Bucharest, Romania. The study was accomplished following the Strengthening the Reporting of Observational Studies in Epidemiology (STROBE) guidelines for observational studies [5]. All data were collected in adherence to the hospital’s standard protocols for managing patients with diabetes. The study was approved by the Ethics Committee for Clinical Studies of the “Nicolae Malaxa” Clinical Hospital, with the approval number 75/2022. All participants involved in the study granted their informed consent for the collection of data and the subsequent utilization of medical information for research objectives.

### 2.2. Study Population

The study consisted of adult patients diagnosed with type 1 (T1DM) or type 2 diabetes mellitus (T2DM) who underwent consultations at the hospital’s inpatient department throughout the study’s duration, agreed to participate, and signed an informed consent form. On the contrary, an age lower than 18 years, the absence of diabetes, pregnancy, and rejection of signing the informed consent form represented exclusion criteria. 

### 2.3. Data Collection

The data were obtained from the medical records of the patients hospitalized during the study period. Data collection was comprehensive and included multiple aspects of the patients’ medical and personal history, including family history of diabetes, duration of diabetes, personal medical history of obesity, hypertension, heart failure, myocardial infarction, stable angina, stroke, hepatic steatosis, dyslipidemia, metabolic syndrome, lower limb amputations, history of behaviors related to health (smoking and alcohol consumption), presence of microvascular diabetes-related complications (chronic kidney disease, peripheral neuropathy, orthostatic hypotension, retinopathy), socio-economic and demographic factors (age, gender, place of residence), clinical measurements (anthropometric indicators), and paraclinical assessment. 

The places of residence were sorted according to urban and rural areas. 

Regarding the health-related behaviors, for smoking assessment, patients were classified as smokers (active or former smokers) and non-smokers, depending on their self-evaluation responses, while for alcohol consumption evaluation, based on the participants’ self-assessment of their drinking habits, they were categorized as drinkers or non-drinkers. 

### 2.4. Clinical Measurements

The following anthropometric indicators were assessed for each participant, including height (cm), weight (kg), waist circumference (cm), hip circumference (cm), and body mass index (kg/m^2^). WC and HC were determined using a measuring tape, following the standard procedures. 

### 2.5. Paraclinical Assessment

The laboratory parameters analyzed were fasting plasmatic glycemia (FPG), glycated hemoglobin (HbA1c) level, serum creatinine, estimated glomerular filtration rate (eGFR), urinary albumin to creatinine ratio (UACR), uric acid, serum urea, aspartate aminotransferase (AST), alanine aminotransferase (ALT), gamma-glutamyl transferase (GGT), total cholesterol (TC), high-density lipoprotein–cholesterol (HDL-c), low-density lipoprotein–cholesterol (LDL-c), triglycerides (TG), and triglyceride–glucose (TyG) index, a surrogate indicator for the evaluation of insulin resistance. 

HbA1c level was determined using the validated high-performance liquid chromatography (HPLC) method [6]; the device used to determine it was produced by BIO-RAD. The batch number for HbA1c was 64573120. The other parameters were determined using spectrophotometry [7] incorporated into Mindray devices. The batch number for biochemistry was 2023071301. 

eGFR was assessed using the 2021 Chronic Kidney Disease Epidemiology Collaboration (CKD-EPI) formula [8]. 

UACR was obtained using spot urine samples, calculated by dividing the total urinary albumin value in milligrams by the creatinine concentration in grams. 

LDL-c levels were derived either through calculation using a formula—TC minus HDL-c minus TG/5—if TG levels were below 400 mg/dL, or through direct laboratory measurement if TG levels exceeded 400 mg/dL [9]. 

TyG was calculated using the following formula: ln [TG (mg/dL) × FPG (mg/dL)/2] [10]. While there are no universally established cutoff values for the TyG index, according to various research studies, we divided it into quartiles by the 25th, the 50th, and the 75th percentiles and treated categorized variables as follows: quartile 1: ≤9.36, quartile 2: 9.37–9.91, quartile 3: 9.92–10.45, and quartile 4: ≥10.46. 

### 2.6. Definitions

Diabetes mellitus was defined according to the ADA (American Diabetes Association) guideline criteria [11]. 

Resting blood pressure was measured using the auscultatory method with the patient in a seated position. Hypertension was characterized by systolic blood pressure (SBP) exceeding 140 mmHg and diastolic blood pressure (DBP) surpassing 90 mmHg, according to the AHA (American Heart Association) guidelines [12], as well as current use of blood pressure-lowering medication or a documented medical history of physician-diagnosed hypertension.

To classify the nutritional status according to BMI (body mass index), we used the definition of the World Health Organization (WHO): underweight was defined by a BMI < 18.5 kg/ m^2^, normal weight by a BMI between 18.5 and 24.9 kg/m^2^, overweight as a BMI between 25 and 29.9 kg m^2^, and obesity by a BMI over 30 kg/m^2^ [13]. 

The diagnosis of hepatic steatosis was made either through imaging techniques, such as an abdominal ultrasound performed during the hospitalization, or a medically recorded history of it. 

Dyslipidemia was diagnosed by laboratory tests of the lipid profile according to present guidelines or current use of lipid-lowering medication [9]. 

Metabolic syndrome was diagnosed using the harmonized definition, with any three of the following criteria: waist circumference >94 cm for males and >80 cm for females, a raised level of triglycerides (TG) ≥ 150 mg/dL, or specific treatment for this lipid abnormality; a reduced level of high-density lipoprotein cholesterol (HDL-c) < 40 mg/dL in men, <50 mg/dL in women, or specific treatment for this lipid abnormality; increased blood pressure, systolic ≥ 130 or diastolic ≥ 85 mmHg, or treatment of previously diagnosed hypertension; and previously diagnosed type 2 diabetes mellitus [14].

The diagnosis of chronic kidney disease was established based on the Kidney Disease Improving Global Outcomes (KDIGO) criteria using the estimated glomerular filtration rate and albuminuria [15].

Eye fundus examination, part of the ophthalmological evaluation, was employed to evaluate diabetic retinopathy following the protocol outlined in The Early Treatment for Diabetic Retinopathy Study [16]. 

The ankle–brachial index test was conducted to evaluate peripheral arterial disease; the test outcome was determined by the ankle-to-brachial blood pressure ratio, and a value below 0.9 was indicative of the diagnosis [11]. 

### 2.7. Study Outcome

The primary endpoint was to assess the clinical and metabolic particularities of a sample Roma population in comparison with a corresponding group of non-Roma patients. 

### 2.8. Statistical Analysis

The IBM SPSS 19th version was used for performing the statistical analysis. The normally distributed continuous variables were presented as mean ± SD (standard deviation) and non-normal variables were reported as median ± IQR (interquartile range), while the categorical variables were presented as absolute counts and percentages. Normality tests which were employed included the Kolmogorov–Smirnov test with a Lilliefors correction for significance and the Shapiro–Wilk statistic [17,18]. For quantitative variables, analysis of variance (ANOVA) was used for comparisons among groups, while for categorical variables, the χ^2^ test was used. For non-normally distributed variables, we applied the Kruskal–Wallis test. Linear regression analysis was performed to identify factors associated with the TyG index. Statistical significance was set at a 95% confidence interval.

## 3. Results

### 3.1. General Characteristics of the Patients

The general characteristics of the patients are summarized in Table 1. The study included 458 Roma patients and 350 Caucasian participants, of which the majority were men (54.6% and 51.5%, respectively). In both groups, most of the patients included had T2DM, reaching 95.1% the Caucasian group and 87.8% among the Roma patients. However, more Roma patients had T1DM (12.2% versus 4.9%). More than half of the participants from both groups did not have a family history of diabetes. 

Regarding the place of residence, irrespective of their ethnicity, most of the patients lived in urban areas (65.1% of the Caucasian patients and 52.2% of the Roma patients). 

A large proportion of Caucasian patients were non-smokers (73.7%) compared to the corresponding group, where the majority of the Roma were smokers (both former or active smokers, 50.4%). Alcohol consumption accounted for 26.9% in the Roma group, while in the Caucasian group, slightly fewer patients (25.1%) were alcohol users (Table 1).

### 3.2. Prevalence of Comorbidities

Regarding the associated diseases, a significant percentage of the Caucasian patients had hypertension (81.1%), approximately 15% more than in the Roma group (67.7%). There were slight differences regarding dyslipidemia, with 78.5% among the non-Roma and 76.7% among the Roma. Apart from this, a predominance of obesity was noticed, with more than half of the patients in both groups categorized as obese (62.2% in the Roma group and 50.3% in the Caucasian group). The prevalence of metabolic syndrome was significantly high among both groups, reaching 94.3% in the Roma patients and 89.1% in the non-Roma (*p* = 0.008) (Figure 1). Hepatic steatosis was present in 55.3% of the Caucasian participants and 48.5% in the Roma subjects (Figure 1).

Analyzing the prevalence of cardiovascular disease, there were no differences concerning personal history of myocardial infarction, which approximately 12% of the patients from each group had; however, the number of patients with a history of stroke was 2.1 times higher in the Roma group compared to non-Roma participants (42 versus 20 patients, *p* = 0.067). The prevalence of stable angina and heart failure was significantly lower in the Caucasian group (*p* = 0.0001). Peripheral artery disease was significantly more frequent in the Caucasian group compared to the Roma (21.7% versus 9.6%). A modest percentage of patients suffered from lower limb amputations, with no differences observed between the two groups (Figure 1).

Detailed data regarding the number of patients with these associated diseases are provided in Appendix A. 

### 3.3. Prevalence of Diabetic Complications

The most prevalent microvascular complication was peripheral polyneuropathy, exceeding 70% in both groups (Figure 2). In the Roma population, chronic kidney disease was present in 22.1% of the patients, while in the Caucasian group, nearly 35% were associated with this complication. Apart from this, almost a third of the participants from both groups were associated with diabetic retinopathy, a slightly higher percentage being observed among the Caucasian patients (38.3% versus 33.25%). Orthostatic hypotension was more prevalent in the Roma population compared to the corresponding group, accounting for 14.6% (Figure 2). 

Comprehensive information on the number of patients with these associated conditions can be found in Appendix A.

### 3.4. Clinical and Paraclinical Assessments

Comparing the Roma population with the Caucasian group, the mean age was lower in the Roma group (55.62 ± 11.55 versus 62.06 ± 10.6 years); moreover, the median duration of diabetes was significantly lower (6.00 versus 11.00 years, *p*-value = 0.0001) (Table 2). 

Analyzing the anthropometric measurements, the average height of the Caucasian patients was slightly higher compared to the opposite group (166.83 ± 9.68 cm versus 164.54 ± 9.27 cm), but the average weight, waist circumference, and hip circumference, however, were higher among the Roma patients (87.51 ± 20.12 kg versus 84.81 ± 17.82 kg, 110 ± 15.87 cm versus 103.73 ± 14.70 cm, and 108.61 ± 14.40 cm versus 104.82 ± 14.02 cm, respectively). Concerning the mean BMI, the same trend was observed (32.28 ± 7.03 kg/m^2^ versus 30.41 ± 5.06 kg/m^2^). 

Regarding the paraclinical assessments, the Roma patients had a slightly higher mean HbA1c level compared to the non-Roma group (9.91 ± 2.45% versus 9.07 ± 2.09%); apart from this, the mean values of the lipid profile were also significantly higher in this ethnic group, with the exception of the mean value of HDL-c, which was higher among the Caucasian patients. Concerning the insulin resistance measured by the TyG index, a slightly higher mean value was observed among the Roma group (10.07 ± 0.71 versus 9.71 ± 0.82). The mean values of the renal profile parameters (creatinine level and uric acid level, respectively) were higher among the Roma patients, but there were no differences regarding the mean urea level. However, the median value of the eGFR level was lower among the Roma patients (80.00 ± 41.00 mL/min/1.73 m^2^ versus 83.00 ± 45.00 mL/min/1.73 m^2^), but the median value of the urinary albumin to creatinine ratio was quite similar (25.00 ± 31.28 mg/g versus 25.13 ± 104.27 mg/g, respectively). The mean values of the hepatic enzymes were higher among the Roma population compared to the corresponding group (Table 2). 

The differences between patients with T1DM and T2DM cand be found in Appendix A. 

### 3.5. Evaluation of the Insulin Resistance among the Study Groups 

Regarding the insulin resistance analyzed using the TyG index, our results revealed that a greater percentage of the patients from the Caucasian group belonged to the first quartile (34%), while most of the Roma patients belonged to the second one (36.9%). Among the fourth quartile, there were significantly more Roma patients than non-Roma patients (24.5% versus 18.6%, *p* < 0.001) (Figure 3). 

Details about the number of patients belonging to each quartile are available in Appendix A.

Concerning the differences between patients with T1DM and T2DM, extended data can be found in Appendix A.

Among patients from the fourth quartile of the TyG index, the mean age of the non-Roma subjects was significantly higher compared to the Roma (62.05 ± 11.67 versus 55.26 ± 11.73, *p* < 0.001); however, the average value of the HbA1c was more elevated among the Roma group (11.66 ± 2.05 versus 9.55 ± 2.36, *p* < 0.001). Regarding the anthropometric measurements, the mean values of weight, WC, and HC were higher in the Roma group compared to the non-Roma group. Concerning the renal profiles, the mean values of the creatinine, eGFR, urea, and uric acid were higher among the Roma, but conversely, the median value of the UACR was more elevated in the Caucasian group of patients. Analyzing the hepatic laboratory tests, the median value of the ALT was higher among the Roma, but with a GGT level which was more elevated among the non-Roma (Table 3). 

### 3.6. Factors Associated with TyG Index

To evaluate the factors associated with the TyG index, we used univariate analysis (Pearson correlation). The statistical analysis is presented in Appendix A.

In non-Roma participants, TyG was positively correlated with weight, HbA1c level, TC, LDL-c, and UACR, while among the Roma group, univariate analysis revealed a positive association between TyG index and HbA1c level, TC, LDL-c, creatinine level, and UACR, as well as a negative association with HDL-c. Multiple linear regression was conducted to estimate the independent correlation of the TyG index with the aforementioned parameters. Variables that were significantly associated with the TyG index among the non-Roma were HbA1c level, TC, HDL-c, and LDL-c (Table 4). In multivariate analysis, variables that were correlated with TyG index among the Roma participants were the HbA1c level and HDL-c (Table 4).

## 4. Discussion

The data that we used are from a Romanian sample of adults among whom T2DM was predominant. We analyzed the clinical and metabolic particularities of a Roma population compared to a corresponding group of non-Roma, including the prevalence of cardiovascular risk factors, cardiovascular disease, health-related behaviors, and anthropometric and paraclinical measurements.

Although T2DM was predominant, interestingly, there were approximately twice as many Roma patients with T1DM compared to non-Roma patients (12.2% versus 4.9%). There is limited specific research on the prevalence of T1DM in the Roma population compared to the general population; therefore, the exact reasons for this higher prevalence could be multifactorial and complex. Further research focusing on the interplay between genetic, environmental, and social factors is needed to fully understand these disparities [19].

Regarding the prevalence of cardiovascular risk factors, the most frequent factors identified in our study were hypertension, obesity, dyslipidemia, smoking, and alcohol consumption. Our findings are comparable with the results from a paper by Enache et al. that analyzed a group of Roma patients from Călărași County, Romania. The prevalence of obesity in our study was likewise higher among the Roma patients; however, greater rates were observed, with approximately 60% in the Roma group and 50% in the Caucasian group versus 45.2% and 43.9%, respectively. Hypertension and dyslipidemia were also significantly prevalent among both groups, but the rates were correspondingly higher among the Caucasian participants [20]. Nevertheless, data from another study that compared Roma patients from Călărași County with the general population showed higher percentages regarding the prevalence of obesity, but even so, the rates were lower than in the results in our paper, with obesity being present in 43.2% of the non-Roma and 43.3% of the Roma. What should be mentioned is that diabetes (known and newly diagnosed) was present in only 10% and 13.6% of the Roma patients, respectively [21]. 

Weiss et al. implied a prevalence of 33% of obesity, 33.62% of hypertension, 26.92% of dyslipidemia, and 42.55% of smoking among the analyzed Roma patients, suggesting lower rates compared to our study; nonetheless, only 15.13% of the patients had diabetes mellitus [22]. 

In the Predatorr study, a representative study in Romania that included more than 2500 patients aged between 20 and 79 years old, the prevalence of obesity was 56.4% in patients with known diabetes and 52.3% in patients with unknown diabetes. Similar rates were observed only among the non-Roma, the prevalence of obesity among the Roma being higher; however, the prevalence of hypertension was around 64% in patients with both known or unknown diabetes, and similar rates were observed only among the Roma. Concerning smoking status, the prevalence was around 50% in patients with both known and unknown diabetes. Our paper identified the same rate among the Roma, and a lower rate was observed among the non-Roma [23].

In analyzing our results in comparison with those from a Roma population in Serbia, among whom the prevalence of previously diagnosed diabetes was 5.9% and newly diagnosed T2DM was 5.2%, we found that only a third of the patients presented a family history of diabetes; our findings, however, suggest a higher percentage of patients with a positive family history of diabetes (45.4%) [24]. 

In Slovakia, a sample from the Roma population had a BMI above 25 kg/m^2^ (55.8% of the Roma men and 53.4% of the Roma women, respectively), while a significant proportion of them had a BMI higher than 30 kg/m^2^ (28.8% of the Roma men and 26.2% of the Roma women, respectively), with a higher mean waist circumference in women than in men [25]. In our study, we analyzed the Roma in comparison with a corresponding group of non-Roma and identified a higher mean BMI value among them (32.28 kg/m^2^ versus 30.4 kg/m^2^), as well as higher mean waist and hip circumferences (110.00 cm versus 103.73 cm, and 108.61 cm versus 104.82 cm, respectively). It would be intriguing to investigate potential disparities between genders.

In Hungary, data from a sample of Roma people implied that, regarding the anthropometric parameters, they had a lower mean height and weight, but with no differences regarding the average BMI compared to the general population; in our study, only the average weight was higher among the Roma group compared to the non-Roma, and concerning the mean BMI, there were also slight differences between the two groups [1]. 

Although the high frequency of communicable diseases among the Roma is firmly highlighted by the current medical literature, data regarding the prevalence of non-communicable diseases, including cardiovascular disease, are still modest. The prevalence of heart disease among the adult Roma population is considered to be around 10% [26]. However, it remains the main cause of premature mortality among this ethnic group, according to a paper from Slovakia [27]. Our study revealed corresponding results: 12% of the Roma participants were associated with myocardial infarction and 9.2% had a history of stroke. Nonetheless, higher rates were observed concerning stable angina and heart failure, with 29.5% of the Roma patients and 19% of them, respectively. 

Regarding the prevalence of metabolic syndrome among the Roma, our study revealed a significant percentage of 94.3%, slightly higher compared to the non-Roma (89.1%). Data from the medical literature, however, identified a prevalence of 36.38% among the Roma individuals in Hungary and 29.28% in Slovakia [1,23,28]. 

Piko et al. analyzed the changes in the prevalence of metabolic syndrome and its components between 2011 and 2018 among a Hungarian sample of the Roma population and identified an increase in the prevalence of central obesity, hypertension, and metabolic syndrome in 2018, as well as an increased TG level. However, in 2018, a reduction in previously diagnosed diabetes mellitus or raised FPG, as well as a reduced HDL-c level, were observed [29]. The increased prevalence of metabolic syndrome among Roma patients is likely due to a combination of genetic predisposition, socio-economic challenges, lifestyle factors, and limited access to healthcare. Tackling these disparities necessitates a comprehensive strategy that involves enhancing healthcare access, encouraging healthier lifestyles, and implementing targeted interventions for this vulnerable population [29,30,31].

Adany et al. analyzed the TyG index among a sample Roma population from Hungary and identified that there were no differences in comparison with a corresponding group from the general Hungarian population (an average value of 4.88 was found in both groups) [32]. In our study, however, a higher mean value of the TyG index was found among the Roma group (10.07 ± 0.71 versus 9.71 ± 0.82). Among the general population, the TyG index has been positively associated with TC, LDL-c, very-low-density lipoprotein–cholesterol (VLDL-c), uric acid, AST, and ALT [33]. Accordingly, in our study, among the non-Roma group, TyG was positively correlated with TC, HDL-c, and LDL-c. The TyG index has been associated with various components of metabolic syndrome, including obesity, dyslipidemia, hypertension, and hyperglycemia [4]. Accordingly, in our study, we observed high mean values of TyG index among both groups, as well as high prevalences of obesity, dyslipidemia, hypertension, and metabolic syndrome.

Concerning the paraclinical examination, a study from Bulgaria revealed higher mean values of the total cholesterol, triglycerides, and LDL-c among a Roma population compared to a non-Roma group, but a higher average HDL-c level among the latter [34]. These results are in concordance with our findings. 

Regarding the prevalence of diabetic microvascular complications among the Roma population, data from the medical literature are scarce. Of the general population with T2DM, chronic kidney disease is considered to affect 25% of the patients, retinopathy is considered to be present in 21% of the patients, and approximately more than 50% of the patients have polyneuropathy [35]. Our study revealed corresponding data regarding the prevalence of chronic kidney disease among the Roma, but with higher rates observed with reference to peripheral neuropathy and retinopathy. A paper by Weiss et al. identified that retinopathy affected 12.5% of the Roma diabetic patients [22].

Regarding the prevalence of amputations, a recent study from Romania suggested that 51.2% of the total number of non-traumatic amputations were in patients with diabetes [36]. Other papers indicated a prevalence of 3.6% in the general Romanian population, while a comparison between non-Roma and Roma patients implied a higher prevalence among the former group (7.5% versus 2.3%) [37,38]. Our study identified a percentage of around 3.5% among both groups.

Overall, considering all results, our study revealed that there are no major differences between the two groups of patients with diabetes mellitus. Apart from this, the results align with those in another recent paper published regarding diabetes distress [39]. These findings are in agreement with the genetic study of Nardos et al., which suggested that genetic predispositions are similar between the Roma and the general population [40]. However, minor differences in clinical and paraclinical assessments and complications in our study could likely be more attributable to disparities in social and living conditions rather than genetic differences, which further emphasizes the importance of addressing social inequalities to improve diabetes management and outcomes in the Roma population. The European Commission and Romanian government have implemented policies to improve Roma health, education, and social services, such as the National Roma Integration Strategy and the EU Framework for National Roma Integration Strategies. While progress has been made, gaps remain in terms of effectively reaching the Roma population. Healthcare providers can bridge the gap by offering culturally competent care and advocating for policy changes. Understanding the Roma’s socio-economic and cultural context is essential for improving patient outcomes and reducing health disparities [41].

## 5. Limitations of the Study

In addressing the constraints of our research, participant recruitment was confined to a single tertiary care hospital amid the COVID-19 pandemic, which may not be representative of the general Roma and non-Roma populations with diabetes, although the sample size is reasonable. Moreover, our study relied on self-reported data for health behaviors like smoking and alcohol consumption, which can be inaccurate due to social desirability bias (people under-reporting unhealthy behaviors).

Apart from this, our study did not collect specific data on healthcare management strategies, such as the frequency of primary healthcare utilization or participation in health education programs among the Roma population. Future studies should aim to include these variables to provide a more comprehensive understanding of healthcare management strategies and their impact on health outcomes within this population. By incorporating these aspects, we can better understand the disparities in healthcare access and outcomes between Roma and non-Roma patients and develop targeted interventions to address these gaps.

While our findings align with larger studies comparing the general population to ethnic minorities, there is a scarcity of data on Roma patients. Consequently, we acknowledge the imperative for additional research in this area, with a more in-depth analysis to identify the specific genetic and environmental factors contributing to the higher cardiovascular risk in the Roma population. Considering conducting a longitudinal study to investigate cause-and-effect relationships between observed factors and health outcomes by partnering with community leaders and healthcare professionals from the Roma community could contribute significantly to improving cardiovascular health outcomes and reducing ethnic disparities in healthcare for the Roma population.

## 6. Conclusions

Taking the above into consideration, the prevalence of cardiovascular risk factors, cardiovascular disease, and microvascular complications among the study Roma population is quite significant, underscoring the importance of considering ethnic disparities in approaching healthcare management strategies. Regarding the anthropometric and paraclinical assessments, our data demonstrated different characteristics of the Roma in comparison to a corresponding group of non-Roma, highlighting the importance of tailoring healthcare interventions to the specific needs and cultural context of this ethnic minority. Further research is warranted to explore the underlying factors contributing to these differences as well as potential genetic or environmental factors, and to develop targeted interventions in high-risk populations.

## Figures and Tables

**Figure 1 biomedicines-12-01422-f001:**
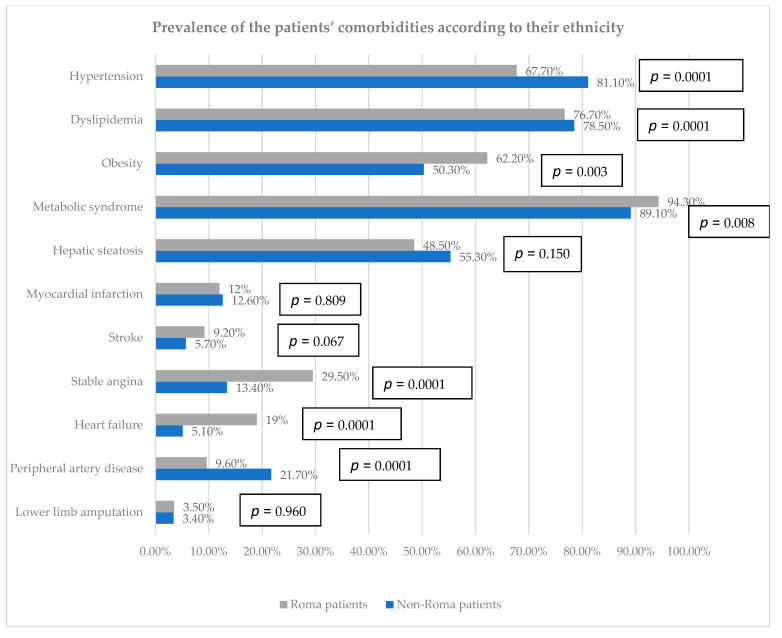
Prevalence of the patients’ comorbidities according to their ethnicity.

**Figure 2 biomedicines-12-01422-f002:**
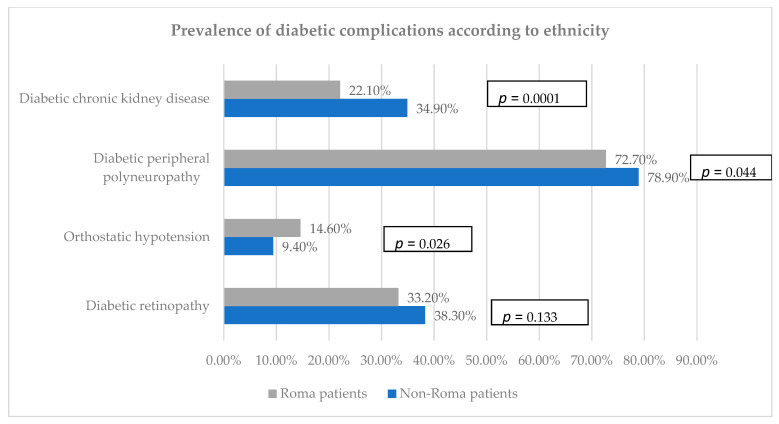
Prevalence of diabetic complications according to ethnicity.

**Figure 3 biomedicines-12-01422-f003:**
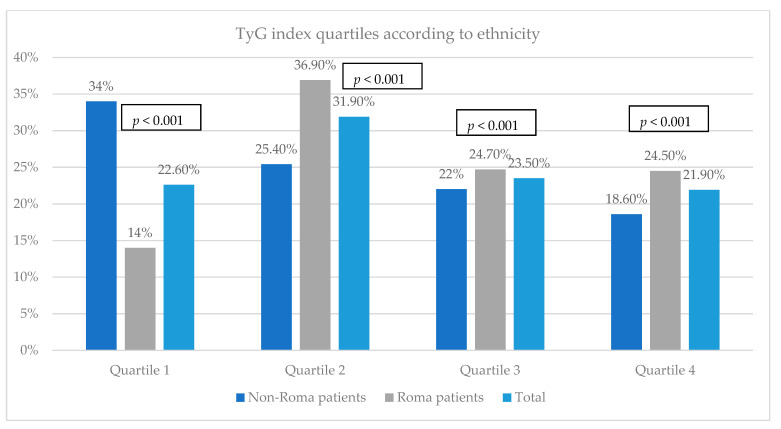
TyG index quartiles according to ethnicity.

**Table 1 biomedicines-12-01422-t001:** General characteristics of the patients.

Variables		Non-Roma Patients (*n* = 350)	Roma Patients (*n* = 458)	*p*-Value
Gender	Men	54.6% (*n* = 191)	51.5% (*n* = 236)	0.391
	Women	45.4% (*n* = 159)	48.5% (*n* = 222)	
Place of residence	Urban area	65.1% (*n* = 228)	52.2% (*n* = 239)	0.0001
	Rural area	34.9% (*n* = 122)	47.8% (*n* = 219)	
Type of diabetes	T1DM	4.9% (*n* = 17)	12.2% (*n* = 56)	0.0001
	T2DM	95.1% (*n* = 333)	87.8% (*n* = 402)	
Family history of diabetes	Yes	47.1% (*n* = 165)	45.4% (*n* = 208)	0.625
Smoking (former or active smokers)	Yes	26.3% (*n* = 92)	50.4% (*n* = 231)	0.0001
Alcohol consumption	Yes	25.1% (*n* = 88)	26.9% (*n* = 123)	0.583

**Table 2 biomedicines-12-01422-t002:** Mean values of the analyzed parameters according to ethnicity.

Parameters	Non-Roma Patients (*n* = 350)	Roma Patients (*n* = 458)	Total (*n* = 808)	*p*-Value
	Mean ± SD	Mean ± SD	Mean ± SD	
Age (years)	62.06 ± 10.6	55.62 ± 11.55	58.41 ± 11.59	0.0001
Duration of diabetes (years)	11.00 ± 8.18 *	6.00 ± 6.89 *	9.00 ± 7.77 *	0.0001
Height (cm)	166.83 ± 9.68	164.54 ± 9.27	165.67 ± 9.54	0.001
Weight (kg)	84.81 ± 17.82	87.51 ± 20.12	86.19 ± 19.06	0.059
WC (cm)	103.73 ± 14.70	110.00 ± 15.87	107.08 ± 15.64	0.0001
HC (cm)	104.82 ± 14.02	108.61 ± 14.40	107.20 ± 14.36	0.011
BMI (kg/m^2^)	30.41 ± 5.60	32.28 ± 7.03	31.36 ± 6.43	0.0001
HbA1c (%)	9.07 ± 2.09	9.91 ± 2.45	9.53 ± 2.33	0.0001
FPG (mg/dL)	226.35 ± 87.96	232.00 ± 117.42 *	243.59 ± 106.46	0.0001
TC (mg/dL)	192.66 ± 65.29	217.12 ± 63.41	205.67 ± 65.40	0.0001
HDL-c (mg/dL)	49.40 ± 14.09	45.57 ± 9.91	47.38 ± 12.20	0.0001
TG (mg/dL)	192.06 ± 138.62	234.39 ± 123.45	214.54 ± 132.39	0.0001
LDL-c (mg/dL)	103.64 ± 48.67	123.11 ± 52.59	113.79 ± 51.64	0.0001
TyG index	9.71 ± 0.82	10.07 ± 0.71	9.90 ± 0.78	0.36
Creatinine (mg/dL)	0.96 ± 0.37	1.04 ± 0.43	1.00 ± 0.40	0.010
eGFR (mL/min/1.73 m^2^)	83.00 ± 45.00 *	80.00 ± 41.00 *	83.00 ± 41.00 *	0.255
Urea (mg/dL)	44.56 ± 20.08	44.34 ± 18.49	44.43 ± 19.16	0.904
Uric acid (mg/dL)	5.99 ± 1.98	6.16 ± 2.36	6.06 ± 2.15	0.578
UACR (mg/g)	25.13 ± 104.27 *	25.00 ± 31.28 *	24.14 ± 65.7 *	0.003
AST (UI/L)	20.00 ± 22.66 *	23.00 ± 25.88 *	22.00 ± 13.18 *	0.086
ALT (UI/L)	24.00 ± 27.31 *	29.00 ± 28.75 *	27.00 ± 23.00 *	0.002
GGT (UI/L)	33.35 ± 102.97 *	44.00 ± 47.58 *	42.00 ± 38.00 *	0.081

Abbreviations: WC (cm)—waist circumference, HC (cm)—hip circumference, BMI (kg/m^2^)—body mass index, HbA1c (%)—glycated hemoglobin, FPG (mg/dL)—fasting plasmatic glycemia, TC (mg/dL)—total cholesterol, HDL-c (mg/dL)—high-density lipoprotein–cholesterol, LDL-c (mg/dL)—low-density lipoprotein–cholesterol, TG (mg/dL)—triglycerides, TyG index—triglyceride–glucose index, eGFR (mL/min/1.73 m^2^)—estimated glomerular filtration rate, UACR (mg/g)—urinary albumin to creatinine ratio, AST (UI/L)—aspartate aminotransferase, ALT (UI/L)—alanine aminotransferase, GGT (UI/L)—gamma-glutamyl transferase. The data are represented as mean ± SD (standard deviation) and median ± IQR (marked with “*”, IQR—interquartile range). The statistical significance was considered at a *p*-value < 0.05.

**Table 3 biomedicines-12-01422-t003:** Characteristics of the patients from the fourth quartile of the TyG index according to ethnicity.

Parameters	Forth Quartile of the TyG index		*p*-Value
	Non-Roma Patients (*n* = 65)	Roma Patients (*n*= 112)	
Age (years)	62.05 ± 11.67	55.26 ± 11.73	<0.001
Duration of diabetes (years)	12.00 ± 4.50 *	6.00 ± 9.30 *	<0.001
Height (cm)	167.03 ± 9.27	164.88 ± 8.27	0.123
Weight (kg)	85.55 ± 17.83	90.64 ± 20.30	0.102
WC (cm)	104.44 ± 15.26	112.65 ± 15.48	0.006
HC (cm)	106.74 ± 8.82	109.07 ± 13.49	0.422
BMI (kg/m^2^)	31.33 ± 4.23	31.83 ± 6.98	0.159
HbA1c (%)	9.55 ± 2.36	11.66 ± 2.05	<0.001
FPG (mg/dL)	255.50 ± 100.50 *	337.50 ± 108.50 *	0.010
TC (mg/dL)	244.28 ± 93.54	247.82 ± 70.12	0.775
HDL-c (mg/dl)	45.44 ± 13.72	42.52 ± 9.71	0.103
TG (mg/dL)	401.25 ± 178.75	363.50 ± 125.93	0.102
LDL-c (mg/dL)	104.35 ± 40.20	127.42 ± 32.80	0.006
Creatinine (mg/dL)	1.02 ± 0.43	1.09 ± 0.43	0.661
eGFR (mL/min/1.73 m^2^)	76.63 ± 29.26	78.16 ± 24.09	0.716
Urea (mg/dL)	42.26 ± 17.36	49.56 ± 19.82	0.500
Uric acid (mg/dL)	5.90 ± 2.13	6.80 ± 2.80	0.363
UACR (mg/g)	45.80 ± 150.08 *	36.00 ± 60.20 *	0.047
AST (UI/L)	40.88 ± 20.00	23.00 ± 15.5 *	0.320
ALT (UI/L)	27.83 ± 20.09 *	31.00 ± 26.25 *	0.904
GGT (UI/L)	94.13 ± 95.00 *	44.00 ± 27.00 *	0.076

Abbreviations: WC (cm)—waist circumference, HC (cm)—hip circumference, BMI (kg/m^2^)—body mass index, HbA1c (%)—glycated hemoglobin, FPG (mg/dL)—fasting plasmatic glycemia, TC (mg/dL)—total cholesterol, HDL-c (mg/dL)—high-density lipoprotein–cholesterol, LDL-c (mg/dL)—low-density lipoprotein–cholesterol, TG (mg/dL)—triglycerides, TyG index—triglyceride–glucose index, eGFR (mL/min/1.73m^2^)—estimated glomerular filtration rate, UACR (mg/g) —urinary albumin to creatinine ratio, AST (UI/L)—aspartate aminotransferase, ALT (UI/L)—alanine aminotransferase, GGT (UI/L)—gamma-glutamyl transferase. The data are represented as mean ± SD (standard deviation) and median ± IQR (marked with “*”, IQR- interquartile range). The statistical significance was considered at a *p*-value < 0.05.

**Table 4 biomedicines-12-01422-t004:** Factors associated with TyG index according to ethnicity.

Variables	B	SE	*p*-Value	95% CI	
				Lower	Upper
Non-Roma patients					
Weight (kg)	−0.001	0.002	0.706	7.215	8.420
HbA1c (%)	0.148	0.018	<0.001	0.112	0.183
TC (mg/dL)	0.015	0.001	<0.001	0.012	0.017
HDL-c (mg/dL)	−0.022	0.003	<0.001	0.028	−0.015
LDL-c (mg/dL)	−0.012	0.002	<0.001	−0.015	−0.009
ALT (UI/L)	0.0007	0.001	0.945	−0.002	0.002
UACR (mg/g)	0.0001	0.0001	0.059	0.0001	0.0001
Roma patients					
HbA1c (%)	0.150	0.027	<0.001	0.096	0.204
TC (mg/dL)	0.004	0.003	0.232	−0.002	0.009
HDL-c (mg/dL)	−0.017	0.008	0.030	−0.032	−0.002
LDL-c (mg/dL)	0.0001	0.004	0.911	−0.008	0.007
Creatinine (mg/dL)	−0.071	0.170	0.678	−0.411	0.269
UACR (mg/g)	0.001	0.001	0.344	−0.001	0.002

Abbreviations: HbA1c (%)—glycated hemoglobin, TC (mg/dL)—total cholesterol, HDL-c (mg/dL)—high-density lipoprotein–cholesterol, LDL-c (mg/dL)—low-density lipoprotein–cholesterol, ALT (UI/L)—alanine aminotransferase, UACR (mg/g)—urinary albumin to creatinine ratio, B—standard beta coefficient, SE—standard error, CI—confidence interval. The statistical significance was considered at a *p*-value < 0.05.

## Data Availability

The data presented in this study are available upon request from the corresponding author. The data are not publicly available due to the hospital’s privacy policy.

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
