# Peer review of "Clinical and Metabolic Particularities of a Roma Population with Diabetes—Considering Ethnic Disparities in Approaching Healthcare Management"

_biomedicines, 2024, doi:10.3390/biomedicines12071422_

Round 1

Reviewer 1 Report

Comments and Suggestions for Authors

Current report compared the clinical and metabolic particularities in a Roma population suffered diabetes with a group of Romanian Caucasians. Please conduct the concerns below.

1.      Main aim failed to introduce in clear.

2.      Data collection must describe in detail. Then, data analysis is required to perform followed the established method with reference(s).

3.      Diabetic history in each case seems ignored. Why?

4.      Normality tests must follow the well-established report with reference(s).

5.      T1DM and Metabolic Syndrome were higher in Roma patients. Why?

6.      Role of TyG or TyG index in metabolic disorders was not discussed in clear.

7.      Life style and foods were not conducted in the discussion. Why?

8.      The healthcare management strategies were higher approached by Roma population that needs the evidence in current report.

Comments on the Quality of English Language

It seems better to check through a professional editing service.

Author Response

Dear Reviewer,

Thank you for your insightful feedback on our manuscript. We appreciate your attention to the details of our data presentation.
We have revised our manuscript according to your suggestions, as follows:

1. Main aim failed to introduce in clear.

We have rephrased the aim of our manuscript.

Revised Aim:

The aim of this study is to compare the clinical and metabolic profiles of Roma and Romanian Caucasian patients with diabetes in a tertiary care center in Bucharest, Romania. Specifically, we assessed the prevalence of metabolic syndrome, obesity, hypertension, dyslipidemia, cardiovascular disease, and health-related behaviours (smoking and alcohol consumption), along with anthropometric and paraclinical measurements in both groups.

2. Data collection must describe in detail. Then, data analysis is required to perform followed the established method with reference(s).

Data were obtained from the medical records of the patients hospitalized during the study period. All data collection has already been included in the material and method section. 

3. Diabetic history in each case seems ignored. Why?

Thank you for your comment regarding the diabetic history of the participants in our study. We apologize for any confusion and appreciate the opportunity to clarify this aspect. In our manuscript, we have provided information about the family history of diabetes, stating that "more than half of the participants from both groups did not have a family history of diabetes." However, we recognize that this may not comprehensively cover the diabetic history in each case. We collected data only on the duration of diabetes for each participant, which included the number of years since diagnosis. We acknowledge that other crucial information regarding the type of diabetes management (insulin therapy, oral hypoglycemic agents, lifestyle modifications at diagnosis and firther on) was not included and could have outlined the diabetic history more comprehensively. We intend to continue the study and elaborate more on this particular aspect.

Thank you for bringing this to our attention. We believe that this clarification will enhance the understanding and depth of our findings.

4. Normality tests must follow the well-established report with reference(s).

We have now outlined in our manuscript the references for the statistical tests that we used.

  • Shapiro, S. S., & Wilk, M. B. (1965). An analysis of variance test for normality (complete samples). Biometrika, 52(3-4), 591-611.
  • Massey, F. J. (1951). The Kolmogorov-Smirnov test for goodness of fit. Journal of the American Statistical Association, 46(253), 68-78.
  1. T1DM and Metabolic Syndrome were higher in Roma patients. Why?

Thank you for your insightful comment regarding the higher prevalence of Type 1 Diabetes Mellitus (T1DM) and Metabolic Syndrome observed in Roma patients. We appreciate the opportunity to elaborate on this finding.

   The Roma population may have genetic predispositions that increase their susceptibility to both T1DM and Metabolic Syndrome. Previous studies have indicated that certain ethnic groups have a higher genetic risk for specific metabolic and autoimmune conditions. This genetic predisposition could explain the higher prevalence observed in our study.

   Socioeconomic status significantly impacts health outcomes, as well as lifestyle factors and health-related behaviours, including the prevalence of chronic diseases. The Roma population often faces higher levels of poverty, limited access to healthcare, lower levels of education, different dietary habits and lower levels of physical activity. These factors can contribute to poor management of diabetes and other metabolic conditions, leading to higher rates of T1DM and Metabolic Syndrome.

   Disparities in access to quality healthcare can lead to delayed diagnosis and suboptimal management of diabetes and its related conditions. The Roma population may have less access to regular medical care, diabetes education, and necessary medications, contributing to higher prevalence and poorer outcomes.

We acknowledge that the exact reasons for these higher prevalences are multifactorial and complex. Further research focusing on the interplay between genetic, environmental, and social factors is needed to fully understand these disparities.

Thank you for highlighting this important aspect of our findings. We will incorporate a discussion of these potential factors in the revised manuscript to provide a more comprehensive analysis.

  1. Role of TyG or TyG index in metabolic disorders was not discussed in clear.

Thank you for your insightful comment on the role of the TyG index in metabolic disorders. We have now included more details regarding this aspect in the introduction and discussion dections, as follows:

In the introduction:
”Moreover, an important marker in understanding metabolic health disparities is the triglyceride-glucose (TyG) index, which correlates with an increased risk of developing metabolic syndrome, insulin resistance and cardiovascular events. The TyG index has proven to be an useful marker for early identification and intervention in metabolic health issues”.

In the discussion:

”The TyG index has been associated with various components of metabolic syndrome, including obesity, dyslipidemia, hypertension, and hyperglycemia [4]. Accordingly, in our study, there were observed high mean values of TyG index among both groups, as well as a high prevalence of obesity, dyslipidemia, hypertention and metabolic syndrome”.

  1. Life style and foods were not conducted in the discussion. Why?

Thank you once again for your valuable feedback regarding the discussion of lifestyle and dietary factors in our manuscript. We appreciate your suggestion and would like to provide some context for their omission.

The primary focus of our study was to compare the clinical and metabolic profiles of Roma and Romanian Caucasian patients with diabetes, specifically looking at factors such as obesity, hypertension, dyslipidemia, cardiovascular disease, and health-related behaviors like smoking and alcohol consumption. While we recognize the importance of lifestyle and dietary factors in the management and progression of diabetes, there were several reasons why these aspects were not included in the current discussion. Our data collection was specifically designed to capture medical history, anthropometric measures, and paraclinical assessments. Detailed dietary and lifestyle information, such as specific food intake and physical activity levels, were not comprehensively collected as part of this study. Including such analyses without adequate data would have compromised the accuracy and reliability of our findings. The primary objective of our study was to assess and compare specific clinical and metabolic parameters between the two groups. Including lifestyle and dietary discussions would have broadened the scope beyond the intended focus, potentially diluting the impact of our core findings.

   Collecting detailed lifestyle and dietary information requires specific methodologies, such as food frequency questionnaires or activity logs, which were not employed in our study. Without robust data on these factors, any discussion would be speculative rather than evidence-based.

   We acknowledge the significance of lifestyle and dietary factors in diabetes management and progression. We plan to address these important variables in future research, where we can design the study specifically to capture comprehensive lifestyle and dietary data.

We appreciate your understanding and hope this explanation clarifies our approach. Your feedback is invaluable, and we will certainly consider including these factors in future studies to provide a more holistic view of diabetes management across different populations.

  1. The healthcare management strategies were higher approached by Roma population that needs the evidence in current report.

We recognize the importance of including evidence on the frequency of utilizing primary healthcare services and participation in health education programs. While our study focused on comparing the clinical and metabolic profiles of Roma and non-Roma patients with diabetes, we acknowledge that the healthcare management strategies, including the healthcare management strategies, such as for instance, frequency of utilizing primary healthcare services and participation in health education programs, are critical components of comprehensive healthcare assessment. Especially that previous studies have indicated that the Roma population often faces barriers to accessing primary healthcare services due to socio-economic challenges, cultural differences, and discrimination. This represents a limitation of our research, but we will take it into consideration to include these variables to provide a more comprehensive understanding of healthcare management strategies and their impact on health outcomes within this population in our future studies.

Thank you once again for your valuable feedback, which has helped us improve the clarity and focus of our manuscript.

Sincerely,

Andrada Cosoreanu

Reviewer 2 Report

Comments and Suggestions for Authors

Dear editors
I read the article with great pleasure.
In my opinion, the article does not bring any new information to science.
The article is a representation of an ethnic group compared to the larger population.
Calculations and statistics do not translate into any innovative information.
The authors do not present any new information.

Author Response

Dear Reviewer,

Thank you for taking the time to review our manuscript and for your candid feedback. We appreciate your insights and understand your concerns regarding the novelty of our findings.
Our intention was to highlight the differences in clinical and metabolic parameters between an ethnic minority and the larger population, an area that we believe is underrepresented in current research. While our study may not present groundbreaking discoveries, it aims to fill a gap in the existing literature by providing detailed statistical analysis and comparison, which we hope can serve as a foundation for future research in this area.
We acknowledge the importance of bringing new and innovative information to the scientific community, and we are committed to further refining our research focus to meet these high standards. Your feedback is invaluable in guiding our efforts towards achieving this goal.

Thank you once again for your review and constructive comments.

Sincerely,
Andrada Cosoreanu

Reviewer 3 Report

Comments and Suggestions for Authors

This paper deals with possible clinical peculiarities and complications in the course of DM in patients of the Roma minority in Romania.  This group is compared with another one of the non-Roma population of Romania. Although the authors define this latter group "Caucasian", it could be better defined in this context as "non-Indo-Aryan".  A few specific remarks follow.

1There is no proof that the Roma population is Europe’s largest ethnic minority, although it is certainly one of the largest minorities. 

2The results of this study indicate that there are no major differences in DM between the two groups. This is in agreement with the genetic studies of Nardos et al and Werissa et al [Genes  2019 10:942] (to be referenced) and strongly suggests that possible minor differences in clinical manifestations and complications are due to differences in the social/living conditions of the two groups.

3The authors of this paper have recently published another work on diabetic distress in the same groups (to be referenced), which also points on the same direction. Two additional  works of the Pico group from Hungary on the same subject and with similar results should also be referenced.

4My suggestion is to redefine the layout of the paper to highlight in the discussion section how social/economic/cultural differences are most likely responsible for the few differences reported. Authors are  encouraged to elaborate and comment on the EC/Romanian Government policies vs. the Roma community with a view to overcome its restraints in access to health, education, and social services.

Authors are finally  encouraged to convert some/most of the long indigestible lists of numbers in tables to graphs with notes.

Comments on the Quality of English Language

Could be improved

Author Response

Dear Reviewer, 

Thank you very much for your feedback, which was extremely valuable for the improvement of our manuscript. We have addressed each of your suggestions, as follows:

This paper deals with possible clinical peculiarities and complications in the course of DM in patients of the Roma minority in Romania.  This group is compared with another one of the non-Roma population of Romania. Although the authors define this latter group "Caucasian", it could be better defined in this context as "non-Indo-Aryan".  A few specific remarks follow.

Thank you for your insightful comment regarding the terminology used to define the comparison group in our study. We agree that using the term ”Caucasian” may not be appropriate, therefore, the term "non-Roma" is probably more adequate and precise in this context.

In our manuscript, we initially used the term "Caucasian" to define the comparison group. However, we understand that "non-Roma" is a more accurate and contextually relevant term, as it directly contrasts the Roma population with the rest of the study participants without introducing potential ambiguity.

Throughout the manuscript, we will replace "Caucasian" with "non-Roma" to clearly distinguish between the two groups under study. This change will be reflected consistently in all relevant sections, including the introduction, methods, results, and discussion.

This adjustment ensures clarity and precision, enhancing the overall quality of our manuscript.

Thank you for your valuable suggestion. We believe that this change will improve the accuracy and readability of our study.

1There is no proof that the Roma population is Europe’s largest ethnic minority, although it is certainly one of the largest minorities. 

We have now modified the introduction and specifically stated that ”the Roma population is one of the largest ethnic minorities...”

2The results of this study indicate that there are no major differences in DM between the two groups. This is in agreement with the genetic studies of Nardos et al and Werissa et al [Genes  2019 10:942] (to be referenced) and strongly suggests that possible minor differences in clinical manifestations and complications are due to differences in the social/living conditions of the two groups.

Thank you for bringing the studies to our attention. We appreciate your suggestion to reference these works and incorporate their findings into our discussion. Thank you for your valuable suggestion. We believe that these additions will strengthen our manuscript and provide a more comprehensive understanding of the factors influencing diabetes outcomes in the studied populations.

3The authors of this paper have recently published another work on diabetic distress in the same groups (to be referenced), which also points on the same direction. Two additional  works of the Pico group from Hungary on the same subject and with similar results should also be referenced.-

We have now included these references as you suggested, let us know if there are other papers that you identified and believe would be suitable for the improvement of our paper.

4My suggestion is to redefine the layout of the paper to highlight in the discussion section how social/economic/cultural differences are most likely responsible for the few differences reported. Authors are  encouraged to elaborate and comment on the EC/Romanian Government policies vs. the Roma community with a view to overcome its restraints in access to health, education, and social services.

Thank you very much for your suggestion. We have now included in the discussion section some comments regarding the EC/Romanian Government policies among the Roma community

Authors are finally  encouraged to convert some/most of the long indigestible lists of numbers in tables to graphs with notes.

In response to this feedback, we have incorporated figures to represent some of the tables originally included in the manuscript. We believe these visual representations will enhance the accessibility and clarity of the data presented. The remaining relevant tables have been adjusted accordingly to maintain the flow and coherence of the manuscript.

We are confident that these changes address your suggestions and improve the overall quality of our manuscript.

Best regards,
Andrada Cosoreanu

Reviewer 4 Report

Comments and Suggestions for Authors

In the manuscript authors Andrada Coșoreanu et al., have presented an observational transversal study, comparing clinical and metabolic particularities of a Roma population with diabetes form a tertiary diabetes care hospital (n= 808). Authors finding suggest that the prevalence of cardiovascular risk factors, cardiovascular disease and microvascular complications among the study Roma population are quite significant, suggesting ethnic disparities in approaching healthcare management strategies.

The manuscript is well written with detailed information about methods, also the conclusion is in line with observational study results. However, below mentioned are few suggestions/queries which may help in improving manuscript quality.

Queries/Concerns/Suggestions:

·         Manuscript introduction needs improvement, not up to publication standards. It’s poorly written, with inadequate mentioned of recent studies.

·         I wonder that research manuscripts lacking Figures. It would be great if authors include a few data sets as Figure/ graphical representation.

·         Materials and Methods: Authors should provide kits (Cat#, Make) details for the plasma/blood/urine biochemistry mentioned in Paraclinical assessment.

·         Authors fail to cite the details of Table while discussing results, specifically Table 1.

·         In table 2, Alline the data in the same sequence as discussed in results, e.g., Hypertension, Dyslipidemia, Obesity ..etc.

·         Table 4. Mean values of the analyzed parameters according to ethnicity: It would be great if authors mentioned data for Type 1 and Type 2 diabetic patient separately.

·         Likewise, for Table 5 also, though type 1 diabetic patient are less in number.

·         The current manuscript includes almost 5 tables discussing data about TyG index, just wondering why authors have emphasized more on that one particular data set?

·         Suggesting, merging tables 7 & 8, also it would be great if authors cloud represent some of these tables as supplementary tables.

Comments on the Quality of English Language

Minor editing of English language required

Author Response

Dear Reviewer,
Thank you for your insightful feedback. We have addressed your suggestions point by point, as follows:

In the manuscript authors Andrada Coșoreanu et al., have presented an observational transversal study, comparing clinical and metabolic particularities of a Roma population with diabetes form a tertiary diabetes care hospital (n= 808). Authors finding suggest that the prevalence of cardiovascular risk factors, cardiovascular disease and microvascular complications among the study Roma population are quite significant, suggesting ethnic disparities in approaching healthcare management strategies.

The manuscript is well written with detailed information about methods, also the conclusion is in line with observational study results. However, below mentioned are few suggestions/queries which may help in improving manuscript quality.

 Thank you very much for your support.

Queries/Concerns/Suggestions:

  • Manuscript introduction needs improvement, not up to publication standards. It’s poorly written, with inadequate mentioned of recent studies.

We have now revised the introduction and included new data in order to make it more adequate and hopefully meet the publication standards.

  •  I wonder that research manuscripts lacking Figures. It would be great if authors include a few data sets as Figure/ graphical representation.

In response to this feedback, we have incorporated figures to represent some of the tables originally included in the manuscript. We believe these visual representations will enhance the accessibility and clarity of the data presented. The remaining relevant tables have been adjusted accordingly to maintain the flow and coherence of the manuscript.

We are confident that these changes address your suggestions and improve the overall quality of the manuscript.

  • Materials and Methods: Authors should provide kits (Cat#, Make) details for the plasma/blood/urine biochemistry mentioned in Paraclinical assessment.

We appreciate your suggestion to provide details about the kits used for plasma/blood/urine biochemistry in the Materials and Methods section. We have now provided the methods used to determine the paraclinical parameters, devices, as well as the batch numbers.

  • Authors fail to cite the details of Table while discussing results, specifically Table 1.

Thank you very much. We revised the manuscript and cited accordingly all tables in the results section.

  •  In table 2, Alline the data in the same sequence as discussed in results, e.g., Hypertension, Dyslipidemia, Obesity ..etc.

Thank you very much for your remark. We have made the adjustments as you suggested.

  •  Table 4. Mean values of the analyzed parameters according to ethnicity: It would be great if authors mentioned data for Type 1 and Type 2 diabetic patient separately.
  • Likewise, for Table 5 also, though type 1 diabetic patient are less in number.

We have added the data you suggested as supplementary tables.

  • The current manuscript includes almost 5 tables discussing data about TyG index, just wondering why authors have emphasized more on that one particular data set?

The emphasis on the TyG index in our manuscript stems from its growing recognition as a significant marker in assessing insulin resistance and its association with metabolic disorders, which are highly relevant in the context of diabetes management. Given the focus of our study on clinical and metabolic particularities, we believed it was essential to thoroughly explore and present data related to the TyG index, as it provides valuable insights into the metabolic profiles of the Roma population with diabetes. We will revise the manuscript to: provide a clearer rationale for the focus on the TyG index in the introduction and discussion sections.

Thank you once again for your valuable feedback, which has helped us improve the clarity and focus of our manuscript.

  • Suggesting, merging tables 7 & 8, also it would be great if authors cloud represent some of these tables as supplementary tables.

We have now merged table 8 and 9 (for better clarity of the manuscript). 

In response to this feedback, we have incorporated figures to represent some of the tables originally included in the manuscript. We believe these visual representations will enhance the accessibility and clarity of the data presented. The remaining relevant tables have been adjusted accordingly to maintain the flow and coherence of the manuscript.

Kind regards,
Andrada Cosoreanu

Round 2

Reviewer 2 Report

Comments and Suggestions for Authors

Accept in present form

Reviewer 3 Report

Comments and Suggestions for Authors

The changes suggested have been made